# Some Unfamiliar Structural Stability Aspects of Unsymmetric Laminated Composite Plates

**DOI:** 10.3390/ma17153856

**Published:** 2024-08-04

**Authors:** Mehdi Bohlooly Fotovat

**Affiliations:** Department of Strength of Materials, Lodz University of Technology, Stefanowskiego 1/15, 90-537 Lodz, Poland; mehdi.bohlooly-fotovat@p.lodz.pl

**Keywords:** post-buckling, second bifurcation, mode jumping, tertiary equilibrium path

## Abstract

It is widely recognized that certain structures, when subjected to static compression, may exhibit a bifurcation point, leading to the potential occurrence of a secondary equilibrium path. Also, there is a tendency of deflection increment without a bifurcation point to occur for imperfect structures. In this paper, some relatively unknown phenomena are investigated. First, it is demonstrated that in some conditions, the linear buckling mode shape may differ from the result of geometrically nonlinear analysis. Second, a mode jumping phenomenon is described as a transition from a secondary equilibrium path to an obscure one as a tertiary equilibrium path or a second bifurcation point. In this regard, some non-square plates with unsymmetric layer arrangements (in the presence of extension–bending coupling) are subjected to a uniaxial in-plane compression. By considering the geometrically linear and nonlinear problems, the bucking modes and post-buckling behaviors, e.g., the out-of-plane displacement of the plate versus the load, are obtained by ANSYS 2023 R1 software. Through a parametric analysis, the possibility of these phenomena is investigated in detail.

## 1. Introduction

For square plates without the effect of extension–bending coupling, the mode shapes based on linear buckling (or so-called eigenvalue solution) and nonlinear post-buckling analysis are the same. For instance, Figure 1 demonstrates the equilibrium path of the square, which is a simply supported plate made of glass-fiber-reinforced polymer (GFRP), with the material properties outlined in [1] and the layer arrangement of [(45/−45)_4_]_T_. The plate is subjected to a uniform uniaxial in-plane compression $F_{ex}$, and the variation in the deflection of the center of the plate $w_{c}$ with respect to the total thickness of the plate h is plotted in this figure. As seen, both analyses (linear eigenvalue and geometrically nonlinear) lead to one half-wave in both in-plane directions. For tracing the nonlinear analysis, the plate with conditions such as imperfections, lateral load, extension–bending coupling, eccentricity of in-plane loads, and so on may tend to show behavior close to the ideal curve. It should be noted that the ideal curve with a bifurcation point is a rare phenomenon, and usually a tendency to this black solid curve exists by using the above conditions. In Figure 1, there are no extension–bending coupling coefficients (i.e., $B_{11}=B_{12}=B_{22}=0$). However, an imperfection with different amplitudes ζ is applied. In the case of initial imperfection, there are interesting studies which have focused on different types of imperfections [2,3,4]. However, the current study is limited to the imperfection of the first linear buckling mode [5]. Another point is that the solution has two curve parts. First, the right curve is the result of a load increment from zero, and the left curve is the result of a load decrement from a large value. However, both curves correspond to the same mode of deflection. These two parts of the solutions are also reported for different structures as truss [6] and general systems [7]. One question that may arise is that is the mode of both linear and nonlinear analysis always the same? In this paper, a special circumstance will be demonstrated where there are possibilities to observe different modes based on different linear and nonlinear analyses.

Concerning the nonlinear analysis of the structures, there is a very rare phenomenon called mode jumping, which is introduced by a limited number of investigators because it usually happens under special circumstances. Among them, Ungureanu et al. [8] presented the possibility of a second bifurcation point for thin-walled steel members with different types of profile sections. Zhang and Murphy [9] worked on the secondary buckling point of the beams by a judicious choice in the beam length. In other words, longer beams have different modes with respect to shorter beams. But at a special length, the possibility of jumping from one mode to another is high. In another case [10], they investigated the effect of a partial elastic foundation on the tertiary equilibrium state of the beams.

Based on different terminology in the literature, mode jumping, second bifurcation point, and tertiary equilibrium path can be significant for those structures for which the conditions of the occurrence of two different modes exist at the same time. However, mode jumping differs from interactive buckling [11,12], where the structure has a mix of more than one mode shape at the same time. In the case of interactive buckling, the beams and columns with I-section profiles have a significant role, where local and global buckling can be seen simultaneously. In particular, the term cellular buckling can be raised in such structures [13,14,15]. Similar behavior has been discovered in other mechanical systems such as rectangular hollow strut [16] and cylindrical shells [17].

In the case of laminated composites, Pirrera et al. [18] and Coburn et al. [19] investigated the possibility of bi-stability and tri-stability in cylindrical and double curved shells, respectively. However, the effect of the extension–bending coupling matrix for unsymmetric laminated composite structures is one of the interesting topics in the field of the structural stability of composite materials. In this regard, Carrera et al. [20] found that unsymmetric and non-square plates may have a special mode shape in the nonlinear range for which the amplitudes of the half-waves are not same. In other words, one may deal with some larger and smaller half-waves in the post-buckling range. Bohlooly Fotovat and Kubiak [21] presented an analytical solution to understand the reason for the non-bifurcation response in the presence of the extension–bending coupling matrix. Now, there is a good opportunity to combine these two studies [20,21] and focus on the mode jumping of non-square plates.

In this paper, two novel aspects of unsymmetric laminated composite plates under uniaxial compressions are presented. First, the effect of the boundary condition on the deflection response of the plate is presented in Section 2. In the case of simply supported boundary conditions, the mode shape of linear buckling analysis (eigenvalue) and post-buckling (geometrically nonlinear) can be different. The reason for and possibility of this difference are explained in Section 3. Second, the plates with different modes (based on linear and nonlinear analyses) have a possibility of jumping to another mode. This is possible in the presence of imperfection. The details of such a mode jumping are explained in Section 4. In all sections, the results of different analyses are obtained from finite element analysis using ANSYS APDL version 2023 R1 software.

## 2. Finite Element Analysis: Set-Up and Solver

To obtain the linear buckling results of ANSYS, the plate is modeled with the shell element. It should be noted that a four-node element with a size of 1 × 1 mm^2^ is selected. This size selection is based on a convergence study. In the geometrically nonlinear analysis, the equilibrium paths are obtained by static analysis. The element size is 5 × 5 mm^2^. In this analysis, an initial imperfection with the shape of the first buckling mode is employed. The solver is the load-control Newton–Raphson method. Due to the absence of any snapping conditions, it is not necessary to apply any path-following techniques [22]. In order to avoid rigid body motions, one edge is considered immovable, and the opposite edge is considered movable and then compressed. A coupled boundary condition is applied for two movable edges in both in-plane directions. This is the reason why mode shapes in the current study have straight edges. However, they were curved in a similar study [20]. A verification study of the current results of ANSYS is reported in our previous similar work [21].

## 3. Unsymmetric Laminated Composite Plates with Simply Supported Edges

According to the classical lamination theory in thin-walled structures, the lack of symmetry with respect to the middle plane results in a behavior of so-called extension–bending coupling [23]. This means that applying in-plane loads can induce deflection in the structure, and conversely, the deflection may influence the in-plane loads. This is due to the nonzero components of the extension–bending coupling matrix. However, this coupling matrix is necessary, but not in a sufficient condition, for plates being deformed before buckling. Another important factor of such a situation is the existence of simply supported boundary conditions. Figure 2 demonstrates a schematic of a two-layer unsymmetric plate [0/90] considering two different kinds of movable boundary conditions (e.g., simply supported and clamped). As seen, the plate is subjected to an external load $F_{ex}$, and the reaction force of a 0-degree layer $F_{in1}$ is higher than 90-degree layer $F_{in2}$. According to the free body diagram of a pin in the simply supported edge (located in the middle plane of the plate), this different value of loads leads to a moment $M_{in}$ which is inserted from the plate to the pin. Since the pin has no resistance to any moment, it starts to rotate. The rotation of the pin leads to the deflection of the plate. In contrast, the free body diagram of the attaching layer in the clamped edge shows that it obliterates the moment very easily (by $M_{re}$), and the plate remains flat during compression (before buckling). Therefore, the next sections of this paper focus on an unsymmetric laminated plate with all simply supported edges.

## 4. Different Modes in Buckling and Post-Buckling of Non-Square Plates

Usually, the result of buckling mode based on a linear analysis (eigenvalue problem) is the same as the result in the geometrically nonlinear analysis. Therefore, a very small amplitude of this linear mode can be employed as an imperfection to be sure that the plate tends to this mode in a nonlinear analysis. However, one exceptional phenomenon can occur in the unsymmetric and non-square laminated composites. This means that there are possibilities to see different modes of deflection based on linear buckling and nonlinear post-buckling analysis (even with the presence of imperfection of linear buckling mode). The reason for this is illustrated in Figure 3. For instance, it is obvious that a GFRP plate with a lay-up arrangement of [(0/90)_4_] and aspect ratio r=2 (i.e., length/width) has a linear buckling mode (or LBM) as (m,n)=(2,1). In other words, two waves and one half-wave will appear in the loading and perpendicular in-plane directions of the plate, respectively. As seen, due to the existence of edge moments on both of the opposite sides of the plate (see Section 2), the results of the nonlinear analysis indicate that the mode can be either (1,1) or (3,1). Because the directions of these moments do not lead to observing an even number of half-waves, the selection of actual mode (between m=1 or 3) lies in the value of the second buckling load. Therefore, the variations in the buckling loads versus the aspect ratio are plotted in Figure 4a. As seen, blue areas correspond to an even number of half-waves in critical buckling mode (e.g., m=2,4,…). This means that these areas are for those non-square plates which are prone to having different modes based on linear and nonlinear analyses. For example, in the first blue area, the critical buckling load (in a range r=1.42~2.42) corresponds to the mode with two half-waves m=2 (in LBM). In this range, the second buckling load plays a vital role. By considering this quantity, the blue area can be divided into two other sub-areas (Figure 4b). First, there is a range as r=1.42~1.72, where the second buckling mode is m=1 and there is another range as r=1.72~2.42 for m=3. Therefore, the deflections of the plates based on the geometrically nonlinear analysis of the red sub-area are like (m,n)=(1,1), and those of the green sub-area are like (m,n)=(3,1), according to Figure 4b. It should be noted that these modes are for a very initial post-buckling range. Then, a question arises: what is the deflection when it is slightly far away (i.e., not very initially)? By conducting many nonlinear studies in ANSYS software, the evidence proved that for a plate with an aspect ratio in the first sub-area (for this case, r=1.42~1.72), the possibility of the occurrence of a tertiary equilibrium path is almost zero. This means that the plate keeps the mode (m,n)=(1,1) and then will be a mixed of (m,n)=(1,1) and (3,1) in a very far post-buckling range [24]. However, the second sub-area is unstable, and the occurrence of a tertiary equilibrium path is possible, which is described in the next section.

## 5. Existence of Tertiary Equilibrium Path

Figure 5 demonstrates the equilibrium paths of the GFRP plates (r=1.5) with lay-up arrangements of [(0/90)_4_]_T_ and [(90/0)_4_]_T_. According to Figure 3b, it is obvious that the linear and nonlinear modes of such an aspect ratio are (m,n)=(2,1) and (1,1), respectively. In Figure 5, there is not any imperfection. In this case, the solution has no two curve parts (see Figure 1) due to the presence of extension–bending coupling. The right curve is the result of [(0/90)_4_]_T_, and the left curve is the result of [(90/0)_4_]_T_. This order of layer arrangement from the bottom to the top is highly effective on the direction of moment (see Figure 2). However, both curves correspond to the same mode of deflection. In addition, a small value of imperfection of the first buckling mode (m,n)=(2,1) will not affect the curves.

Now it is time to select a plate with an aspect ratio higher than 1.72 (see green area in Figure 4b). The post-buckling response of the GFRP plate with r=2 is demonstrated in Figure 6. In this figure, there is no imperfection and, as mentioned previously, the linear buckling mode is (m,n)=(2,1) and the deflection in post-buckling has a shape like (m,n)=(3,1). However, this post-buckling curve is unstable, and by adding a small value of imperfection of the first buckling mode, the results will be different.

In this regard, the post-buckling curves of the perfect and imperfect plate are plotted in Figure 7a. In this figure, a plate with r=2.2 and different amplitudes of imperfection of the first buckling mode is analyzed. As seen, the second bifurcation point, tertiary equilibrium path, or jumping mode (based on different terminology in the literature) can occur by applying an amplitude of imperfection larger than 0.006. Also, the moment of separation of different curves is magnified in this figure. The differences between the equilibrium paths of a plate with imperfections of 0.006 and 0.008 can be interpreted as confrontation of two quantities. Both parameters (extension–bending coupling and imperfection) force the plate to have a different mode. When the plate has higher imperfection, it overcomes the extension–bending coupling and the mode shape of plate is the same as the mode shape of imperfection. However, a plate with a lower value of imperfection has a mode shape based on a condition that the extension–bending coupling forces on the plate (see Figure 3d).

The counterpart of Figure 7a for the deflection of another point as d with (x,y)=(a/4,b/2) is plotted in Figure 7b. In this case, the curve corresponding to imperfection ζ/h=0.006 shows well that the plate tries to adopt mode jumping. However, due to the very small value of imperfection, the moments of extension–bending coupling return the plate to the mode of second buckling load.

Figure 8 presents the different aspect ratios (e.g., r=1.9,2.0, 2.1, and 2.2) to illustrate the time of mode jumping. In other words, the time of mode jumping means the location of the point where the red path separates from the blue path. As seen, the location of this point is delayed by increasing the value of the aspect ratio. The physical meaning of this delay comes from Figure 4b. According to the green sub-area of Figure 4b, by increasing the aspect ratio from r=1.72 to 2.42, the second buckling load (with the same mode in blue path) is decreased, and it comes closer to the first buckling load (with the same mode in red path). This means that the second buckling mode is becoming stable, and the time of mode jumping from the second to first buckling mode will be postponed.

## 6. Conclusions

In this paper, two linear and geometrically nonlinear analyses of ANSYS software are used and two new topics are presented. (1) A laminated composite plate may have different mode shapes of linear buckling and nonlinear post-buckling analysis. (2) Such a structure may have a behavior called mode jumping in the post-buckling response. In order to observe different modes, the plate should have the following conditions:
(a)Nonzero extension–bending coupling: These couplings, i.e., $B_{11}$ and $B_{22}$, are present for some unsymmetric laminated composite plates. The plate with cross ply lamination, i.e., [0/90]_n_ (n = 2, 4, 8, …), is one of the practical examples.(b)Simply supported boundary conditions: The edges should have no resistance against rotation. It is like a simple pin or free edge without any moment reactions. In the current results, a simple pin or so-called simply supported boundary conditions are investigated.(c)No eccentricity of load and boundary conditions: The resultant of in-plane compressions (through the thickness) and locations of pins should coincide in the middle plane. One of the practical cases is the uniform distribution of compression with which the resultant will coincide in the middle plane.(d)Aspect ratio: The length of the plate should be larger than the width to have an even number of half-waves in the first linear buckling mode. For GFRP material and a layer arrangement of [(0/90)_4_], the aspect ratio should be higher than 1.4.

In the case of the presence of mode jumping, the plate should meet all the above conditions, and the two following cases should be present:(a)Unstable second buckling mode: The number of half-waves in the second linear buckling mode should be higher than those in the first linear buckling mode. For example, a GFRP plate with [(0/90)_4_] and an aspect ratio of 2.1 has first and second buckling mode shapes as (2,1) and (3,1), respectively. So, such a plate has potential to have mode jumping in the post-buckling response.(b)Imperfection: An amplitude of the imperfection of the first buckling mode should be applied as an initial deflection of the plate. However, it should be a bit larger to overcome the effects of the extension–bending coupling.

If the rectangular plate meets all the above conditions, the extension–bending coupling makes the plate have a mode which is the same as the second buckling mode, and the initial imperfection makes the plate have a mode which is same as the first buckling mode. However, the plate will select a mode shape with a lower value of strain energy, which is the first buckling mode. Therefore, the plate will jump to this mode in the post-buckling response. However, the time of jumping is highly dependent on the value of the aspect ratio. For the last result of the current study, it is concluded that the higher value of the aspect ratio has a severe conflict between imperfection and extension–bending coupling. This is because both modes require strain energy in the same range (both modes are approximately stable). This will result in mode jumping with a longer delay in the response.

## Figures and Tables

**Figure 1 materials-17-03856-f001:**
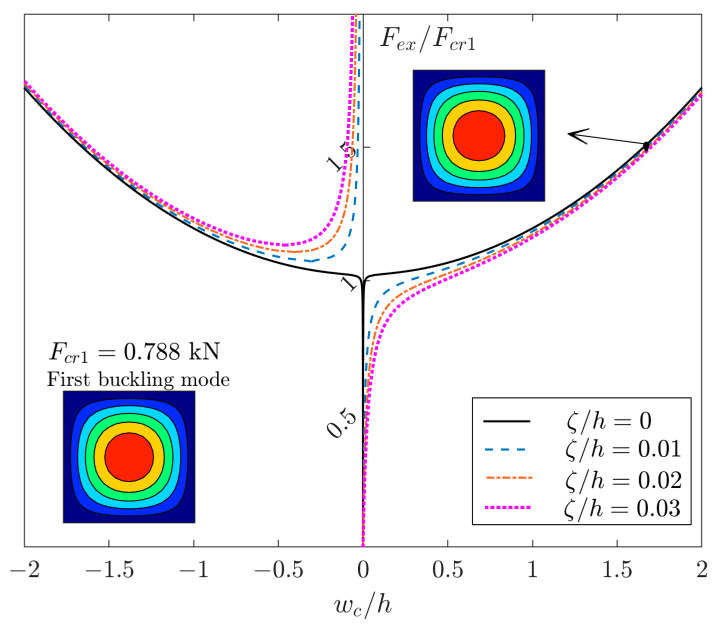
The post-buckling curves of a square GFRP plate [(45/−45)_4_]_T_ with different amplitudes of imperfections.

**Figure 2 materials-17-03856-f002:**
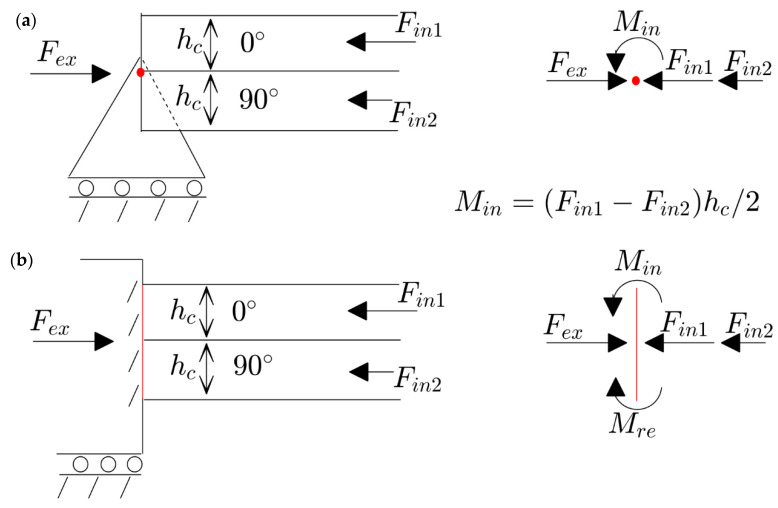
The schematics of reaction forces in a two-layer plate [0/90] with different kinds of boundary conditions and their free body diagrams: (**a**) movable simply support; (**b**) movable clamped.

**Figure 3 materials-17-03856-f003:**
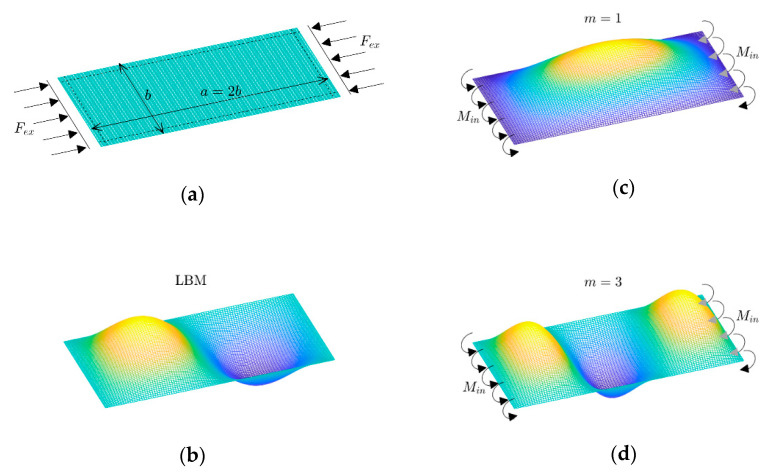
(**a**) A rectangular plate (r=2) under compression, (**b**) linear buckling mode (LBM), and (**c**), (**d**) possible deflections in nonlinear post-buckling analysis (either m=1 or 3).

**Figure 4 materials-17-03856-f004:**
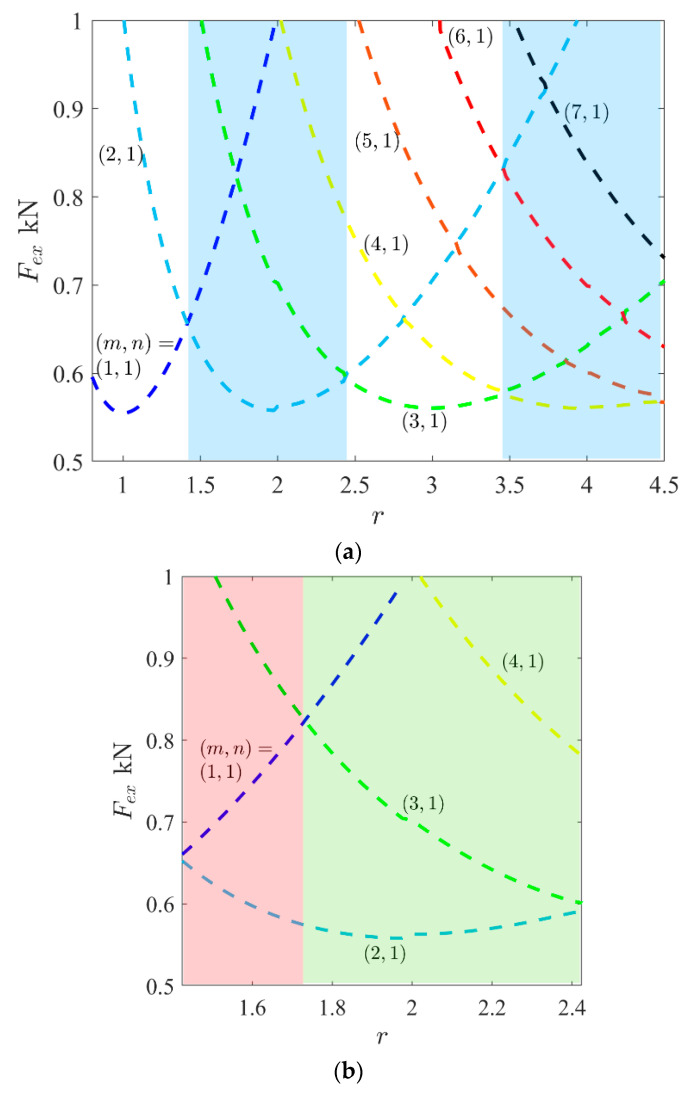
(**a**) The buckling loads of a GFRP plate [(0/90)_4_] under compression versus different aspect ratios. (**b**) The first blue area in detail with two sub-areas.

**Figure 5 materials-17-03856-f005:**
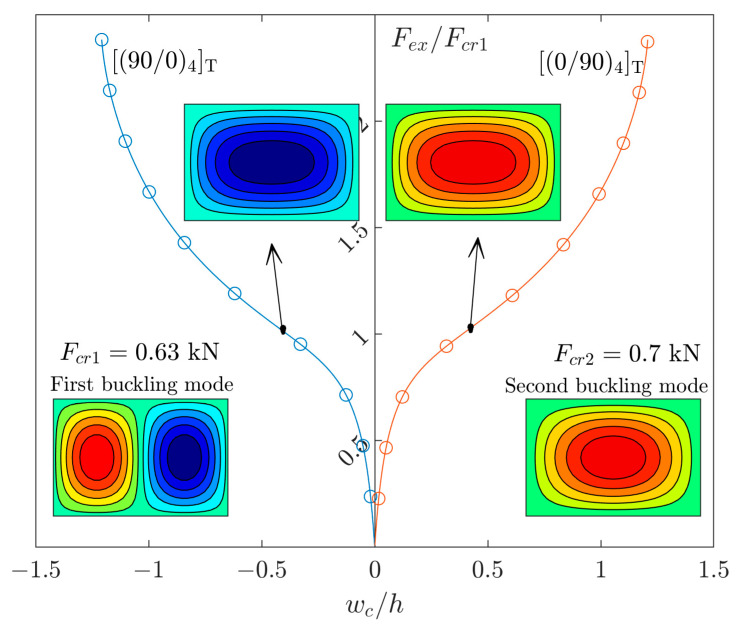
The post-buckling curves of perfect GFRP plates with r=1.5.

**Figure 6 materials-17-03856-f006:**
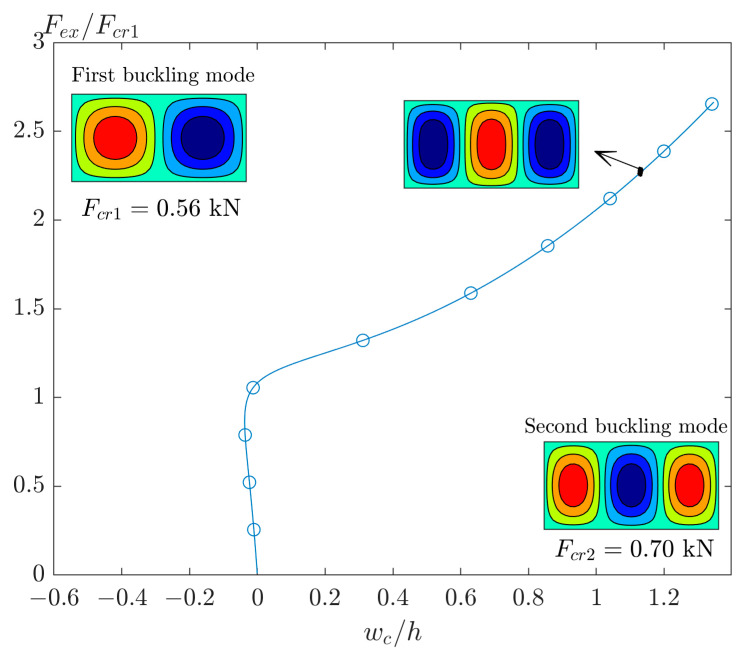
The post-buckling curve of perfect GFRP plate [(0/90)_4_]_T_ with r=2.

**Figure 7 materials-17-03856-f007:**
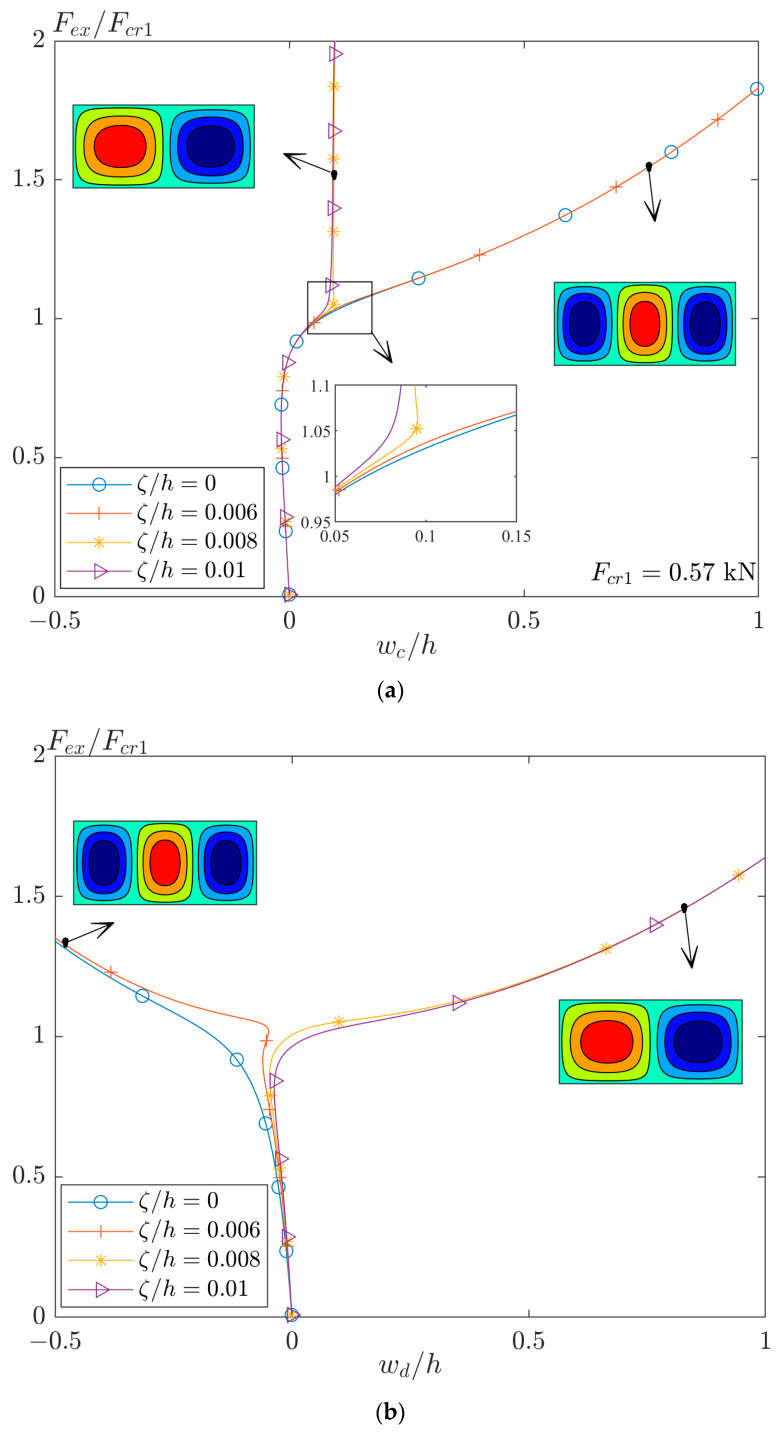
The post-buckling curves of perfect and imperfect GFRP plates [(0/90)_4_]_T_ with r=2.2, (**a**) for displacement at center (x,y)=(a/2,b/2) and (**b**) at (x,y)=(a/4,b/2).

**Figure 8 materials-17-03856-f008:**
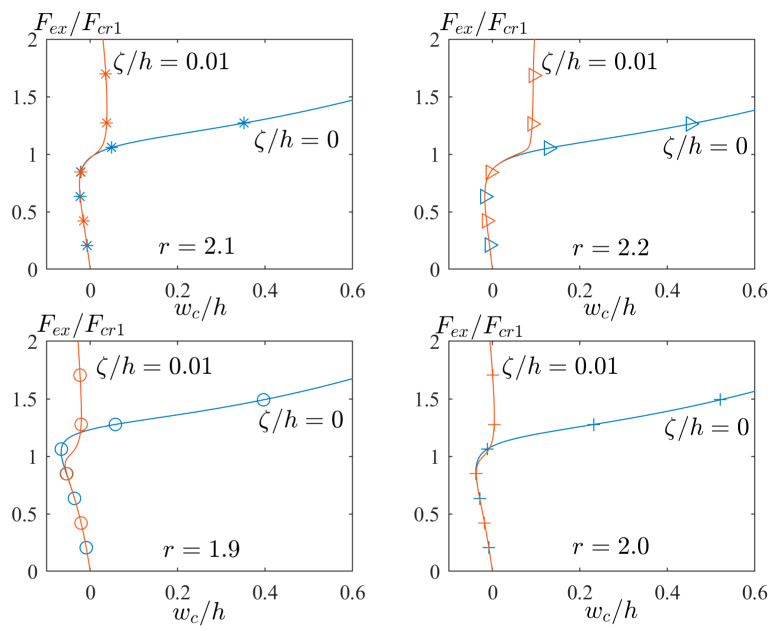
The post-buckling curves of perfect/imperfect GFRP plates with different aspect ratios.

## Data Availability

Data are contained within the article.

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
