# Peer review of "Some Unfamiliar Structural Stability Aspects of Unsymmetric Laminated Composite Plates"

_materials, 2024, doi:10.3390/ma17153856_

Round 1

Reviewer 1 Report

Comments and Suggestions for Authors

The manuscript presents interesting observations on the behavior of unsymmetric laminated composite plates under compression. The topic aligns well with the journal's scope and will likely be of interest to researchers in the field. To further strengthen the manuscript, the authors should address the following points in their revision:

1. Introduction section.

Expand the Literature Review: While the introduction mentions mode jump and multi-stability, it could benefit from a more comprehensive review of existing research.

  • Include relevant work on thin-walled or slender structural systems exhibiting multi-stable states and potential mode jumps.
  • Consider referencing the provided literature on mode jumps in both metal and composite structures
  • Reduce Bias: Ensure the introduction presents a balanced perspective, acknowledging relevant prior work.

 The authors might refer to the following work:

* Mode jump and progression in thin-walled metal structures:

a) Becque, J., & Rasmussen, K. J. (2009). Experimental investigation of the interaction of local and overall buckling of stainless steel I-columns. Journal of structural engineering, 135(11), 1340-1348.

b) Wadee, M. A., & Gardner, L. (2012). Cellular buckling from mode interaction in I-beams under uniform bending. Proceedings of the Royal Society A: Mathematical, Physical and Engineering Sciences, 468(2137), 245-268.

c) Wadee, M. A., & Bai, L. (2014). Cellular buckling in I-section struts. Thin-Walled Structures, 81, 89-100.

d) Shen, J., & Wadee, M. A. (2018). Length effects on interactive buckling in thin-walled rectangular hollow section struts. Thin-Walled Structures, 128, 152-170.

e) Groh, R. M. J., & Pirrera, A. (2019). On the role of localizations in buckling of axially compressed cylinders. Proceedings of the Royal Society A, 475(2224), 20190006.

* Mode jump for thin-walled composites structures:

a) Pirrera, A., Avitabile, D., & Weaver, P. M. (2012). On the thermally induced bistability of composite cylindrical shells for morphing structures. International Journal of Solids and Structures49(5), 685-700.

b) Coburn, B. H., Pirrera, A., Weaver, P. M., & Vidoli, S. (2013). Tristability of an orthotropic doubly curved shell. Composite Structures96, 446-454.

2. Initial Geometric Imperfections.

Acknowledge the Impact: The manuscript should acknowledge the potential influence of initial geometric imperfections on the load-carrying capacity and post-buckling behavior, particularly for structures with multiple stable post-buckling modes.

Reference Suggestions: Consider including references that explore the effects of initial imperfections on post-buckling behavior. The authors might refer to the following work:

* Effects of initial imperfections on the post-buckling behaviour and their programmability.

a) Cox, B. S., Groh, R. M. J., Avitabile, D., & Pirrera, A. (2018). Modal nudging in nonlinear elasticity: Tailoring the elastic post-buckling behaviour of engineering structures. Journal of the Mechanics and Physics of Solids116, 135-149.

b) Shen, J., Pirrera, A., & Groh, R. M. (2022). Building blocks that govern spontaneous and programmed pattern formation in pre-compressed bilayers. Proceedings of the Royal Society A478(2265), 20220173.

c) Shen, J., Garrad, M., Zhang, Q., Leao, O., Pirrera, A., & Groh, R. M. (2023). Active reconfiguration of multistable metamaterials for linear locomotion. Physical Review B107(21), 214103.

3. Nonlinear solver details.

* Specify Solver Type: The author should provide more details about the nonlinear solver they adopted to trace the nonlinear equilibrium path. Clarify whether it's an arclength solver or a normal Newton solver. The information should be clear and specific enough for the potential readers to reproduce the results.

* Stability Verification: Discuss the limitations of ANSYS in detecting negative eigenvalues of the tangential stiffness matrix at converged states. These negative eigenvalues might indicate unstable or non-physical solutions. In ABAQUS, the users can check the output .msg file to check the number of negative eigenvalues at each converged solution. The author is encouraged to see whether they can also get similar information in ANSYS to ensure that their presented results are stable and physically meaningful.

The authors might refer to the following work for further information about the key concepts in path-following:

a) Groh, R. M., Avitabile, D., & Pirrera, A. (2018). Generalised path-following for well-behaved nonlinear structures. Computer Methods in Applied Mechanics and Engineering331, 394-426.

b) Shen, J., Groh, R. M. J., Schenk, M., & Pirrera, A. (2021). Experimental path-following of equilibria using Newton’s method. Part II: applications and outlook. International Journal of Solids and Structures213, 25-40. 

Comments on the Quality of English Language

The writing is generally good and clear. The authors could use AI-tool, such as Google Bard or ChatGPT to further polish their writing. 

Author Response

Authors’ Response to Editor and Reviewers:

The author would like to thank the editor and reviewers, whose comments on the paper helped to improve the investigation on the mode jumping of composite plates. The author addressed all reviewers’ suggestions and clarified some questions. In the revised manuscript, I use yellow highlighting to identify changes made to address comments of the reviewers.

Reviewer #1:

  1. Introduction section.

Expand the Literature Review: While the introduction mentions mode jump and multi-stability, it could benefit from a more comprehensive review of existing research.

  • Include relevant work on thin-walled or slender structural systems exhibiting multi-stable states and potential mode jumps.
  • Consider referencing the provided literature on mode jumps in both metal and composite structures
  • Reduce Bias: Ensure the introduction presents a balanced perspective, acknowledging relevant prior work.

 The authors might refer to the following work:

* Mode jump and progression in thin-walled metal structures:

  1. a) Becque, J., & Rasmussen, K. J. (2009). Experimental investigation of the interaction of local and overall buckling of stainless steel I-columns. Journal of structural engineering, 135(11), 1340-1348.
  2. b) Wadee, M. A., & Gardner, L. (2012). Cellular buckling from mode interaction in I-beams under uniform bending. Proceedings of the Royal Society A: Mathematical, Physical and Engineering Sciences, 468(2137), 245-268.
  3. c) Wadee, M. A., & Bai, L. (2014). Cellular buckling in I-section struts. Thin-Walled Structures, 81, 89-100.
  4. d) Shen, J., & Wadee, M. A. (2018). Length effects on interactive buckling in thin-walled rectangular hollow section struts. Thin-Walled Structures, 128, 152-170.
  5. e) Groh, R. M. J., & Pirrera, A. (2019). On the role of localizations in buckling of axially compressed cylinders. Proceedings of the Royal Society A, 475(2224), 20190006.

* Mode jump for thin-walled composites structures:

  1. a) Pirrera, A., Avitabile, D., & Weaver, P. M. (2012). On the thermally induced bistability of composite cylindrical shells for morphing structures. International Journal of Solids and Structures49(5), 685-700.
  2. b) Coburn, B. H., Pirrera, A., Weaver, P. M., & Vidoli, S. (2013). Tristability of an orthotropic doubly curved shell. Composite Structures96, 446-454.

Response: Thanks for evaluating the current work. Considering this comment, some paragraphs with above references are added to the first section of the revised manuscript.

  1. Initial Geometric Imperfections.

Acknowledge the Impact: The manuscript should acknowledge the potential influence of initial geometric imperfections on the load-carrying capacity and post-buckling behavior, particularly for structures with multiple stable post-buckling modes.

Reference Suggestions: Consider including references that explore the effects of initial imperfections on post-buckling behavior. The authors might refer to the following work:

* Effects of initial imperfections on the post-buckling behaviour and their programmability.

  1. a) Cox, B. S., Groh, R. M. J., Avitabile, D., & Pirrera, A. (2018). Modal nudging in nonlinear elasticity: Tailoring the elastic post-buckling behaviour of engineering structures. Journal of the Mechanics and Physics of Solids, 116, 135-149.
  2. b) Shen, J., Pirrera, A., & Groh, R. M. (2022). Building blocks that govern spontaneous and programmed pattern formation in pre-compressed bilayers. Proceedings of the Royal Society A, 478(2265), 20220173.
  3. c) Shen, J., Garrad, M., Zhang, Q., Leao, O., Pirrera, A., & Groh, R. M. (2023). Active reconfiguration of multistable metamaterials for linear locomotion. Physical Review B, 107(21), 214103.

Response: According to this comment, a short description is added to the first section of the revised manuscript. This description is about the effect of imperfection on the response with citing the references.

  1. Nonlinear solver details.

* Specify Solver Type: The author should provide more details about the nonlinear solver they adopted to trace the nonlinear equilibrium path. Clarify whether it's an arclength solver or a normal Newton solver. The information should be clear and specific enough for the potential readers to reproduce the results.

* Stability Verification: Discuss the limitations of ANSYS in detecting negative eigenvalues of the tangential stiffness matrix at converged states. These negative eigenvalues might indicate unstable or non-physical solutions. In ABAQUS, the users can check the output .msg file to check the number of negative eigenvalues at each converged solution. The author is encouraged to see whether they can also get similar information in ANSYS to ensure that their presented results are stable and physically meaningful.

The authors might refer to the following work for further information about the key concepts in path-following:

  1. a) Groh, R. M., Avitabile, D., & Pirrera, A. (2018). Generalised path-following for well-behaved nonlinear structures. Computer Methods in Applied Mechanics and Engineering331, 394-426.
  2. b) Shen, J., Groh, R. M. J., Schenk, M., & Pirrera, A. (2021). Experimental path-following of equilibria using Newton’s method. Part II: applications and outlook. International Journal of Solids and Structures213, 25-40.

Response: According to this comment, a detailed explanation is added to the new section in the revised manuscript. This explanation is about the setting of solution in the current FEM results of ANSYS.

Kind Regards

Mehdi Bohlooly Fotovat

Lodz University of Technology

Reviewer 2 Report

Comments and Suggestions for Authors

Deformation of laminated composite under compression was numerically discussed using buckling and post-buckling analysis. 

In the post-buckling analysis, the mode changes with the boundary conditions, which generate asymmetry not to occur buckling. In the buckling analysis, we cannot tell which mode will occur, but the possible mode can be obtained. On the other hand, in the post-buckling analysis, we can tell which mode will occur when the boundary condition is given. There is no uncertainty as to which mode will occur. Therefore, the phenomenon itself is not unfamiliar. The expression "jump" is also misleading. This is because every time the same mode will be observed if the boundary condition is the same, which is totally different from buckling in which we cannot tell which mode will occur. The mode changes if the boundary condition changes. That all.

Under near mode-changing conditions, the process may behave slightly differently, but this arises from physical reasons and not from uncertainty. Therefore, when discussing such behavior, why and how need to be clarified.

In the case of r=2.2, different mode was observed for 0.006 and 0.008. Why did these differences arise? What happens if the value is closer to the transition value? The physical explanation of why the behavior goes to the larger mode should be added, not just reporting the calculated results.

In addition, there was no discussion on accuracy of calculation. Especially when boundary conditions are close to unstable conditions, the results may vary depending on the calculation accuracy. 

Author Response

Authors’ Response to Editor and Reviewers:

The author would like to thank the editor and reviewers, whose comments on the paper helped to improve the investigation on the mode jumping of composite plates. The author addressed all reviewers’ suggestions and clarified some questions. In the revised manuscript, I use yellow highlighting to identify changes made to address comments of the reviewers.

Reviewer #2:

  1. In the post-buckling analysis, the mode changes with the boundary conditions, which generate asymmetry not to occur buckling. In the buckling analysis, we cannot tell which mode will occur, but the possible mode can be obtained. On the other hand, in the post-buckling analysis, we can tell which mode will occur when the boundary condition is given. There is no uncertainty as to which mode will occur. Therefore, the phenomenon itself is not unfamiliar. The expression "jump" is also misleading. This is because every time the same mode will be observed if the boundary condition is the same, which is totally different from buckling in which we cannot tell which mode will occur. The mode changes if the boundary condition changes. That all.

Response: Thanks for evaluating the current work. In the results of current study, the boundary conditions are fixed as simply supported at four edges of the plate. However, due to the nonzero extension-bending coupling of laminated composite, there are possibilities to have a mode change during the loading. This possibility is a case of uncertainty, which it is focused to explain why and when it will be happened. The terminology “jump” came from the literature, which investigators usually use mode jumping, second bifurcation, and/or tertiary equilibrium path. According to this comment, a suitable text is added to highlight the meaning of jumping in the first section of the revised manuscript.

  1. Under near mode-changing conditions, the process may behave slightly differently, but this arises from physical reasons and not from uncertainty. Therefore, when discussing such behavior, why and how need to be clarified.

Response: Considering this comment, some paragraphs are added to the revised manuscript to highlight the reasons from physical point of view.

  1. In the case of r=2.2, different mode was observed for 0.006 and 0.008. Why did these differences arise? What happens if the value is closer to the transition value? The physical explanation of why the behavior goes to the larger mode should be added, not just reporting the calculated results.

Response: These differences are highly dependent on the value of initial imperfection, where it can be interpreted as: both parameters (extension-bending coupling and imperfection) force the plate to have a different mode. When the plate has higher imperfection, it overcomes to extension-bending coupling. According to this comment, the interpretation of this figure is improved in the revised manuscript.

  1. In addition, there was no discussion on accuracy of calculation. Especially when boundary conditions are close to unstable conditions, the results may vary depending on the calculation accuracy. 

Response: Considering this comment, a paragraph is added to the revised manuscript, and it is mentioned that this study is the continuation of our past works based on FEM results of ANSYS. All verifications are discussed in those  papers which are cited in the revised manuscript.

Kind Regards

Mehdi Bohlooly Fotovat

Lodz University of Technology

Reviewer 3 Report

Comments and Suggestions for Authors

General Comment

The submitted manuscript presents a numerical research, based on nonlinear finite element analysis, on the buckling and geometrical nonlinear behavior of unsymmetrical laminated composites plates under compression. In particular, and following the findings from previous studies from the author, two novel aspects are presented and discussed: for simply supported boundary conditions, the mode shape of linear buckling (eigenvalue problem) and post-buckling (geometrically nonlinear problem) can be different; and plates with different modes (based on linear and nonlinear analyses) show the possibility to jump to another mode in the presence of imperfections.

After a small literature review, the author presents the results from the numerical study carried out to demonstrate the referred phenomena. The results are discussed and explained.

The manuscript deals with a very interesting topic, related with the stability of thin laminated composite plates. Nowadays, these elements are widely used in several industries, such as aeronautic and automotive. When subjected to compression, stability usually is conditioning for the design. Since this kind of elements allows for stable states after buckling to be found, the findings of new phenomena at this stage turns to be very important for optimal designs.

I consider that the manuscript needs a revision before it can be accepted for publication. I made some comments/suggestions to improve the manuscript (please see comments for the authors). The author should take the suggestions into account, revise its manuscript and resubmit it.

Specific Comment 1

Please add the word “stability” in the title to specify better the aim of the study

Specific Comment 2

The reading must be improved throughout the entire document. Several sentences must be rewritten for better understanding.

Specific Comment 3

The literature review is somewhat scarce and should be expanded, and the references should be updated. Only twelve references exist and three are from the author.

Specific Comment 4

For the sake of the readers, please include a section describing the used FE model (geometry, mesh density and justification, boundary conditions, family of used FE, how the load was applied, validation/calibration of the model, …)

Specific Comment 5

Some subfigures are don’t identified with (a), (b), …

Specific Comment 6

Fig. 8 should be discussed in the text.

Specific Comment 7

The Conclusions section must be improved. After an introductory paragraph, please summarize the main findings in a point by point format. Also, in the end, please add a paragraph and briefly discuss the implications of your findings for practical design.

Comments on the Quality of English Language

Plase see specific comment 2 for the author

Author Response

Authors’ Response to Editor and Reviewers:

The author would like to thank the editor and reviewers, whose comments on the paper helped to improve the investigation on the mode jumping of composite plates. The author addressed all reviewers’ suggestions and clarified some questions. In the revised manuscript, I use yellow highlighting to identify changes made to address comments of the reviewers.

Reviewer #3:

Specific Comment 1

Please add the word “stability” in the title to specify better the aim of the study

Response: According to this comment, the title is changed in the revised manuscript.

Specific Comment 2

The reading must be improved throughout the entire document. Several sentences must be rewritten for better understanding.

Response: According to this comment, the grammar of the English language has been improved significantly. Please let us know if any improvements are required.

Specific Comment 3

The literature review is somewhat scarce and should be expanded, and the references should be updated. Only twelve references exist and three are from the author.

Response: According to this comment, a detailed description is added to the first section of the revised manuscript. This description is about the effect of imperfection and other mode jumping studies.

Specific Comment 4

For the sake of the readers, please include a section describing the used FE model (geometry, mesh density and justification, boundary conditions, family of used FE, how the load was applied, validation/calibration of the model, …)

Response: According to this comment, a new section is added to the revised manuscript. This description is about all conditions and setting of solution in FEM results of ANSYS.

Specific Comment 5

Some subfigures are don’t identified with (a), (b), …

Response: According to this comment, all subfigures are identified in the revised manuscript.

Specific Comment 6

Fig. 8 should be discussed in the text.

Response: According to this comment, a short description is added to text of the revised manuscript. This description is about the figure 8.

Specific Comment 7

The Conclusions section must be improved. After an introductory paragraph, please summarize the main findings in a point by point format. Also, in the end, please add a paragraph and briefly discuss the implications of your findings for practical design.

Response: According to this comment, the conclusion section is improved in the revised manuscript.

Kind Regards

Mehdi Bohlooly Fotovat

Lodz University of Technology

Round 2

Reviewer 2 Report

Comments and Suggestions for Authors

Although the mode jumps dealt with are long-standing and well-known phenomena and do not represent anything scientifically novel, the manuscript has been revised to explain the phenomena more clearly. In the modified part, there was an explanation that the timing of the mode jumping was delayed by increasing the value of the aspect ratio, it is quite difficult to understand what the author would like to insist just from Figure 8. Indeed, the load at the inflection point decreases with increasing r, leading to early buckling. Delays need to be discussed quantitatively.

Author Response

Comment 1: Although the mode jumps dealt with are long-standing and well-known phenomena and do not represent anything scientifically novel, the manuscript has been revised to explain the phenomena more clearly. In the modified part, there was an explanation that the timing of the mode jumping was delayed by increasing the value of the aspect ratio, it is quite difficult to understand what the author would like to insist just from Figure 8. Indeed, the load at the inflection point decreases with increasing r, leading to early buckling. Delays need to be discussed quantitatively.

Response 1: Thanks again for evaluating this work. In this figure, I mentioned about the time of separation. This separation means that the location of the point which red path separates from blue path. As seen, this location for a plate with r=2.2 is occured in a far blue path. In opposite side, this location for a plate with r=1.9 is occured in a early point of blue path. The reason is that the blue path is being more stable when r=2.2 (as seen in Fig.4b).

According to this comment, I tried to clarify my purpose in the revised manuscript. 

Reviewer 3 Report

Comments and Suggestions for Authors

I received the revised version of the article with revised title “Some unfamiliar structural stability aspects of unsymmetric laminated composite plates”. The author has improved the article according to all my previous comments. Hence, I consider that the article can be accepted for publication in the present form.

Author Response

Comment 1: I received the revised version of the article with revised title “Some unfamiliar structural stability aspects of unsymmetric laminated composite plates”. The author has improved the article according to all my previous comments. Hence, I consider that the article can be accepted for publication in the present form.

Response 1: Thank you very much.